# Autonomous Improvement of Instruction Following Skills via Foundation Models

**Zhiyuan Zhou**\*, **Pranav Atreya**\*, **Abraham Lee, Homer Walke, Oier Mees, Sergey Levine**
UC Berkeley

https://auto-improvement.github.io

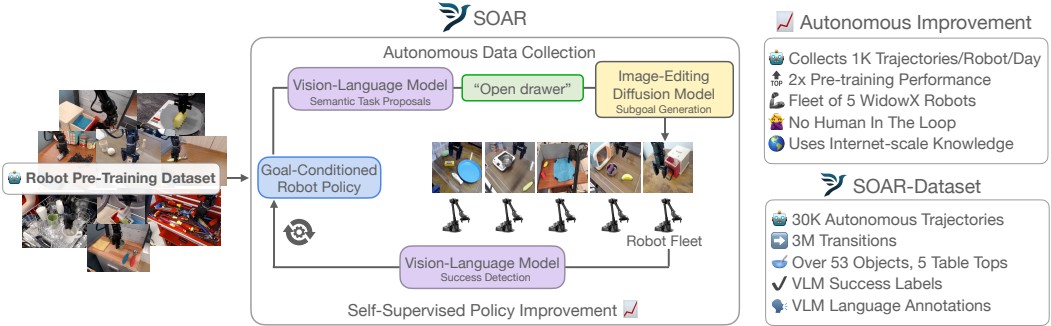

Figure 1: We introduce SOAR, an approach to autonomously improve instruction following policies by leveraging foundation models for large-scale autonomous data collection and self-improvement with no human in the loop. SOAR imports Internet-scale knowledge from pre-trained Vision-Language Models and image-editing Diffusion Models to guide autonomous data collection, and enables policy-improvement with a self-supervised objective on the autonomous data. We make available the 30.5K autonomous trajectories collected with SOAR in SOAR-dataset.

**Abstract:** Intelligent instruction-following robots capable of improving from autonomously collected experience have the potential to transform robot learning: instead of collecting costly teleoperated demonstration data, large-scale deployment of fleets of robots can quickly collect larger quantities of autonomous data that can collectively improve their performance. However, autonomous improvement requires solving two key problems: (i) fully automating a scalable data collection procedure that can collect diverse and semantically meaningful robot data and (ii) learning from non-optimal, autonomous data with no human annotations. To this end, we propose a novel approach that addresses these challenges, allowing instruction-following policies to improve from autonomously collected data without human supervision. Our framework leverages vision-language models to collect and evaluate semantically meaningful experiences in new environments, and then utilizes a decomposition of instruction following tasks into (semantic) language-conditioned image generation and (non-semantic) goal reaching, which makes it significantly more practical to improve from this autonomously collected data without any human annotations. We carry out extensive experiments in the real world to demonstrate the effectiveness of our approach, and find that in a suite of unseen environments, the robot policy can be improved 2x with autonomously collected data. We open-source the code for our semantic autonomous improvement pipeline, as well as our autonomous dataset of 30.5K trajectories collected across five tabletop environments.

## 1 Introduction

Key to the success of modern machine learning methods is the ability to leverage large amounts of weakly labeled data: from scraping the web for free-form text to train large language models [1, 2, 3, 4] to self-supervised training of visual representations on diverse images [5, 6, 7, 8, 9], methods

---

\*Equal contribution. Correspondence to zhiyuan_zhou@berkeley.edu, pranavatreya@berkeley.edu

8th Conference on Robot Learning (CoRL 2024), Munich, Germany.

that can make effective use of larger, more weakly labeled, and more loosely curated datasets tend to exhibit better robustness and generalization. For embodied agents such as robots, following this recipe presents a major challenge: while text, images, and videos can be sourced from the web, there is no existing repository of abundant robot data. While there have been efforts aimed at creating larger robotic datasets [10], it is hard to match the volume and diversity of Internet-scale data, as robot data needs to be collected in the laboratory with costly human effort [11, 12, 13, 14, 15]. A more scalable recipe for robotic data acquisition would be to acquire data *autonomously*, with robots interacting on their own with real-world scenes and objects, subject to minimal human supervision. However, building a robot learning system to collect and effectively make use of such autonomous data under realistic conditions requires addressing a number of technical challenges: deciding which tasks to collect data for [16, 17], designing a self-supervised robotic learning analogue to the scalable methods employed in other fields, such as NLP and CV [1, 3, 5], and ensuring that the autonomous self-improvement process is stable and requires minimal human intervention [18, 19, 20, 21].

In this work, we explore a simple idea to overcome these challenges: while human-provided robotic demonstration data is costly to collect at scale, we can leverage cheap Internet-scale data to learn about semantics, and cheap autonomous experience to learn about physics, robot motion, and environment interaction, connecting these two sources of experience through vision-language models.

This allows us to improve robotic instruction-following policies without costly additional human supervision, while focusing on tasks that are semantically meaningful and connecting the robot's behaviors to language instructions. To instantiate this idea, we present a complete robotic system that starts with a set of behaviors learned from previously collected offline data, and then improves its repertoire of skills through autonomous data collection guided by task proposals from a vision-language model (VLM) [22], effectively importing Internet-scale knowledge. To enable the robot to improve its ability to follow language instructions from autonomous experience, we separate the instruction following system into a semantic component, which interprets language commands and converts them into subgoal images, and a functional component, which then attempts to reach these goal images. The semantic component can be instantiated as a language-conditioned image generation model trained on Internet-scale data [7, 23], and the functional component can be instantiated as a goal-conditioned policy [24] trained on unlabeled robot data, without any additional supervision. Finally, a VLM scores the success of the executed behaviors in reaching the proposed tasks. In this way, all of the parts of our system that require understanding semantics – task proposals, scoring and instruction following – can acquire these semantic concepts from Internet-scale pre-training, while all of the parts that require controlling a robot can utilize unlabeled, autonomously collected data.

Figure 1 highlights our proposed approach and its benefits. We introduce a semantics-aware autonomous system designed to improve skills without human intervention, using natural language to index semantic skills. The process starts with VLM that generates task proposals by drawing on its broad understanding of environment affordances. For example, it will only suggest opening a drawer if the drawer is currently closed. To learn from autonomously collected data, the system must link each trajectory to the semantic meaning of the task completed. This becomes challenging when semantics are expressed through natural language, as the robot only receives one bit of feedback per trajectory: whether or not it successfully completed the language-based task. This becomes even more challenging when the success labels for trajectories might occasionally be incorrect, such as those generated by a VLM. To address this, we use a particular form of instruction following policy that decouples language understanding from robotic control. Specifically, we train a *goal*-conditioned policy, re-framing the semantic concepts as goal images instead of language instructions. We command the goal-conditioned policy to follow language tasks by synthesizing image subgoals from language via an image-editing diffusion model [23]. This allows us to formulate learning as a goal-conditioned problem, with dense supervision provided through hindsight relabeled image goals [25]. This decoupling also allows for a high level of generalization. The image subgoal generator, pre-trained on internet data, performs well in environments with distribution shifts—precisely the kind of environments we aim to improve autonomously. Finally, to determine whether the robot successfully achieved the commanded language instructions during autonomous rollouts, we again use a VLM to label each trajectory, focusing policy improvement on semantically meaningful tasks.

As the main contribution of our work, we propose **S**caled **C**ollection for **A**utonomous Imp**r**ovement (**SOAR**), a general-purpose robotic system for autonomous improvement of multi-task language-conditioned policies in varied real-world environments. While our system makes use of a number of

components developed in prior work, it combines them in a novel way to enable self-improvement of general-purpose robotic policies, and to our knowledge is the first to demonstrate a self-improving robotic policy that does not rely on human hand-specified downstream tasks or fine-tuning on additional demonstrations. We deployed SOAR on a fleet of five WidowX robot arms in various real-world scenes experiencing distribution shifts. During the few weeks of SOAR's deployment, we collected over $30,000$ trajectories (totaling $3M$ transitions) of autonomous data. We demonstrate that SOAR can effectively utilize this data to autonomously improve policy performance by 2x across 10 different scenes.

## 2 Related Work

**Instruction following robot policies.** Language is a natural interface for instructing robots and there is significant prior work on training language-conditioned policies [26, 27, 28, 29, 30, 31], transferring language understanding from pre-trained Large Language Models (LLMs) and VLMs [32, 33, 34, 35, 36, 37, 38, 39, 40, 41], and using language to decompose long-horizon tasks [42, 43, 44, 45, 46]. Our focus is on improving a language-conditioned policy with autonomous data, by decoupling language from motion skills. We decompose skills into a language-conditioned subgoal image generator and a goal-conditioned policy. In contrast with directly conditioning actions on language inputs, we observe that motion skills can be improved in a self-supervised way, using goal-conditioning as self-supervision [17, 47, 48, 49, 50, 51, 52, 53, 54]: the policy is trained to reach its hindsight goals [25] and can thus learn from sub-optimal trajectories. We use an image-editing diffusion model to produce goals based on language, as in the SuSIE method [23], together with goal-conditioned behavioral cloning (GCBC) to produce a *decomposed* language-conditioned policy and use it to autonomously improve instruction-following robot policies.

**Autonomous improvement of robot policies.** Recently, the most capable robot policies have leveraged largely optimal and static demonstration datasets with imitation learning or offline reinforcement learning (RL) [55, 28, 10, 56, 57, 58, 59, 17, 60, 61, 62, 63, 64, 65]. Our work focuses on the *autonomous improvement* problem: we use prior knowledge of high-level semantics and low-level skills obtained from large-scale pre-training to collect data on a fleet of robots and improve instruction-following policies. Some prior work has explored training policies from scratch purely from autonomously collected data with self-supervised learning [66, 65, 67, 68, 69] or online RL [70, 16, 71, 72, 73]. Others have attempted to improve pre-trained robot policies with autonomous data using either conditional behavior cloning [61, 74] or RL [75, 19, 58, 16, 71, 76, 77, 78]. However, these methods are limited in that they do not improve language-conditioned skills and they require additional human demonstrations to bootstrap self-improvement. For instance, Bousmalis et al. [61] autonomously improve a goal-conditioned policy rather than improving language-conditioned skills, and they rely on 500 to 1000 human demonstrations of each improvement task when fine-tuning the policy, training task-specific reward functions, and obtaining goal images for commanding the policy. Yang et al. [19] autonomously improve language-conditioned skills, however they similarly use human demonstrations of the improvement tasks for pre-training the policy and fine-tuning task-specific VLM-based reward functions. Kalashnikov et al. [16] and Kumar et al. [58] improve the performance of task-index conditioned rather than language-conditioned skills, and they rely on hand-collected success examples to train a reward classifier. In comparison, our system is designed to improve language-conditioned skills in an entirely self-supervised manner. We use a frozen VLM to both choose language-conditioned skills to improve and filter the self-collected data for successes.

**Autonomous data collection and task proposals.** To improve an instruction-following policy, we use a vision-language model (VLM) trained on Internet-scale data to propose semantically meaningful tasks given the current state of the robot's environment. The use of Internet-scale pre-trained models to propose tasks has also been explored by Xian et al. [79] and Wang et al. [80] in the context of simulated environments. Most similar to our setup is AutoRT [81], which automatically proposed tasks using a VLM and LLM: the VLM generates text descriptions of the scene, and the LLM uses the descriptions to generate language tasks. However, AutoRT only focuses on proposing tasks for data collection, without provisions for automatic success detection, goals, or other components necessarily for improvement of robotic policies. In contrast, our work describes a complete autonomous self-improvement cycle, where the proposed tasks are used to automatically collect data, continually improving a language-conditioned policy.

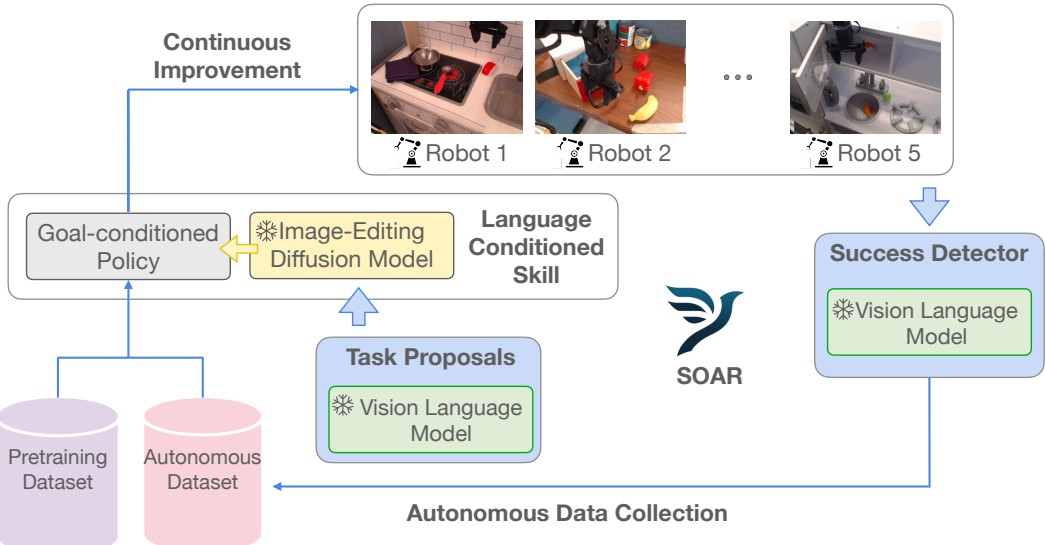

Figure 2: Overview of the SOAR autonomous improvement pipeline: First, we equip the robot with a set of basic skill by pre-training. Then, we deploy the pre-trained policy on a fleet of five robots to autonomously collect data, with a VLM proposing viable language tasks to practice. Specifically, the language task is turned into a subgoal image via an pre-trained image-editing diffusion model, and the robot executes a goal-conditioned policy. Finally, we use a VLM to label success information of the collected trajectories, and train the policy using this data, resulting in improvement.

## 3  SOAR

In this section, we present our general-purpose system for autonomously enhancing multi-task instruction-following policies in real-world environments. SOAR comprises four components that collectively facilitate this autonomous improvement. Fig. 2 illustrates our semantics-aware autonomous improvement pipeline: after training the robot on a pre-training dataset to equip it with a set of basic skills, the pre-trained policy is deployed across a fleet of robots to gather data using automated task proposals and success detection. Finally, we retrain the policy with the autonomously collected data to achieve further improvement.

**Component 1: VLM task proposals.** When deploying language-based instruction-following policies in real-world scenes, it is crucial for the collected autonomous experience to involve meaningful interactions that manipulate the world. This ensures that the policy improves at tasks that humans find valuable. In theory, this can be achieved by querying a capable VLM for task proposals, which takes in an image of the environment and outputs a language task. However, we find that our chosen VLM, CogVLM [22], is not yet sophisticated enough to fully reason about the intersection between the environment's affordances and the robot's physical capabilities (e.g., it might suggest opening a microwave door that is out of the robot's reach). Nonetheless, if we pre-specify a list of tasks the robot can physically perform, CogVLM is adept at reasoning about spatial relationships and environment affordances. More detailed information can be found in Apendix B.1.1.

When the VLM finds that multiple plausible tasks can be meaningfully commanded, we pick the task that maximizes diversity of task execution. Formally, given a set of candidate task commands $T = \{\tau_1, \tau_2, ..., \tau_k\}$, we formulate the problem of picking which task to command as a multi-armed bandit problem where the goal of the task-selection agent is to minimize its uncertainty of each task's success rates. We can use the Upper Confidence Bound (UCB) algorithm [82] for this, picking the task according to

$$\tau_{\text{command}} = \underset{\tau_i \in T_{\text{feasible}}}{\arg\max} \sqrt{\frac{\log(N+1)}{n(\tau_i)+1}}, \tag{1}$$

$$\text{and } T_{\text{feasible}} = \{\tau_i : VLM(s, \tau_i) = \text{ feasible}\}. \tag{2}$$

$T_{\text{feasible}}$ is the subset of tasks from $T = \{\tau_1, \tau_2, ..., \tau_k\}$ that the VLM considers feasible to accomplish in state $s$. $n(\tau_i)$ is the number of times task $i$ has been attempted during data collection, and $N = \sum_{\tau_i \in T_{\text{feasible}}} n(\tau_i)$ is the total number of all feasible tasks attempted.

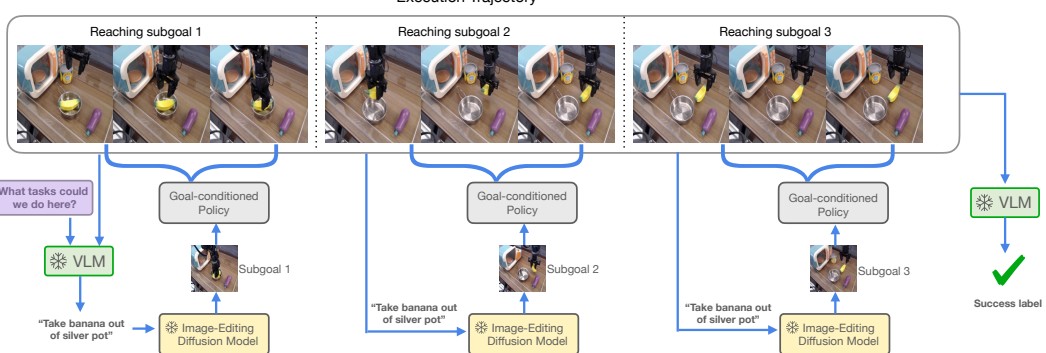

Figure 3: Trajectory rollout with decomposed language-conditioned policy: The VLM proposes a meaningful language task given the observation of the environment, and this language instruction is used to generate a subgoal image via an image-editing diffusion model. The goal-conditioned policy then tries to achieve a sequence of 5 subgoals (only 3 visualized here) each with 20 steps. Finally, the VLM determines if the language instruction has been achieved at the end of a trajectory.

**Component 2: language-conditioned control as goal-conditioned control.** In the context of autonomous improvement, there are two key desiderata any choice of policy must satisfy. Firstly, the policy must exhibit a high-degree of generalizability to handle out-of-distribution environments. Secondly, the improvement algorithm for the policy should be self-supervised, so it can leverage autonomous data without human supervision. Language-conditioned behavior cloning (LCBC) does not meet these criteria [55]: LCBC policies suffer from grounding problems when queried with vocabulary outside the training data [12], and it is hard for a LCBC policy to improve from autonomous robot data without near-perfect language annotations.

We propose instantiating our language-instruction following policy as a foundation model wrapper around a goal-conditioned (GCBC) policy, the foundation model being SuSIE [23], a text-conditioned image-editing diffusion model trained on Internet-scale data and fine-tuned on robotic data. Rather than directly being used for conditioning the policy, the language instruction along with the current observation is sent to SuSIE to generate a subgoal image making progress towards the language task, and this subgoal image is then fed into the GCBC policy. This formulation satisfies our two desiderata. First, it improves the quality of autonomous data collection due to SuSIE's Internet pre-training, with Section 4 demonstrating that generalization capabilities are a crucial advantage of this modular language-conditioned policy. Second, it simplifies learning from autonomous data, as the goal-conditioned training objective allows for more supervision to be extracted from unlabeled and sub-optimal data, which is experimentally validated in Section 4.

Inference with this modular instruction-following policy is depicted in Figure 3. Given the VLM supplied language instruction and the current observation of the robot's environment, SuSIE generates an image subgoal corresponding to the language instruction. This subgoal, along with the current observation of the environment, is fed to the goal-conditioned policy for a fixed 20 timesteps. After this, SuSIE is queried again for the next subgoal. The stochastic nature of the diffusion sampling process means that SuSIE-generated subgoals may vary for the same image observation and language task. Empirically, this variability is beneficial for exploration, as subtly modifying the goal image is akin to adding exploration noise to policy rollouts, with the advantage that the noise is goal-directed. The training procedure of this policy is described in component 4.

**Component 3: VLM success detection.** To enhance the robot's ability to follow instructions, we need a method to identify the parts of the autonomous data where the robot's actions align with the commanded semantic task. Since successful trajectories are more likely to contain meaningful interaction data, we use the VLM to automatically detect success. The VLM receives the language task and the final frame of a trajectory, then classifies whether the trajectory successfully completed the task or not. In SOAR's improvement process, we only re-train on the successful trajectories (according to the VLM) to focus on improving semantically relevant skills. As a baseline we also tested using these success-labeled trajectories to improve a LCBC policy (see Section 4), but found it much less efficient.

**Component 4: policy improvement.** The final component of SOAR is a method for self-supervised policy improvement. The goal is to use the autonomous data to self-improve over the

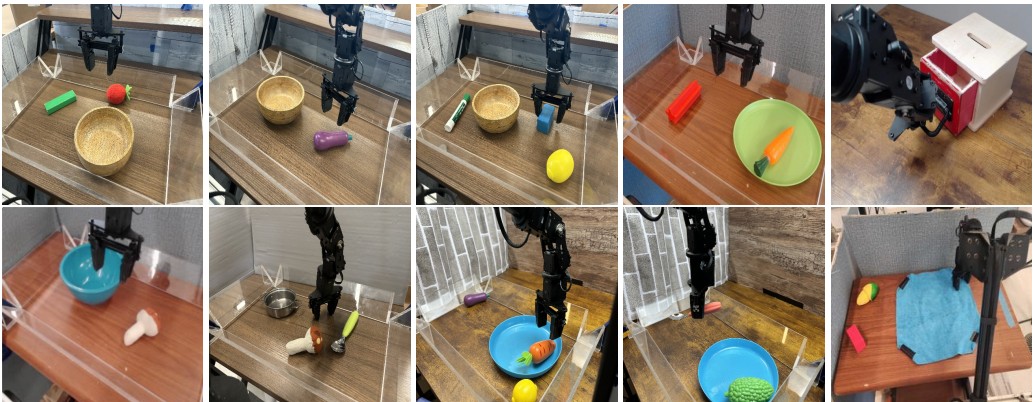

Figure 4: Across the five unique robot workspaces for the five WidowX robots, there are 10 different scenes, each scene corresponding to a distinct set of manipulatable objects available, and each scene supporting many different tasks that can be executed. 8 scenes support pick-and-place tasks, 1 supports drawer opening and closing, and 1 supports deformable cloth manipulation.

pre-trained policy that is used for data collection. Recall that the language-conditioned policy in SOAR is decomposed into a high-level language-to-goal generator and a low-level goal-conditioned policy. Since the low-level control policy is decoupled from language understanding, policy improvement amounts to improving the low-level goal-conditioned policy. As is common in goal-conditioned policy learning, we can learn from hindsight experience [25]: we relabel a portion of training goals as future states actually achieved in the same trajectory during data collection. This objective is particularly appealing in the context of an autonomous improvement setup because goal-conditioned learning from hindsight relabeled goals is a source of self-supervision. We instantiate improvement of the goal-conditioned policy as goal-conditioned behavior cloning (GCBC) on successful autonomous trajectories, where the success determination was made by the VLM. While GCBC is a principled method to learn from failure data as well [83], filtering with the VLM allows us to focus on improving goal-reaching tasks relevant for the semantic skills we care about.

The subgoal diffusion model SuSIE is not updated during data collection and policy-improvement, nor is any component directly tied to language understanding (the VLM for task proposal and success detection is also frozen). This modular formulation of our language-conditioned policy allows self-supervised learning objectives to be aligned with semantic instruction-following improvement, a property unattainable by methods that directly condition on language, such as LCBC.

## 4 Experimental Results

Our experiments aim to evaluate the end-to-end improvement attained by our method, compare our approach to alternative methods, and evaluate the individual design decisions. Specifically, we aim to answer the following research questions:

1. Can SOAR effectively produce autonomous improvement over an initial pre-trained policy?

2. Does decomposing instruction-following skills into a language-conditioned subgoal generation and image goal-conditioned control bring about better policy improvement than directly conditioning the low-level policy on language?

3. Can SOAR autonomously propose meaningful tasks and collect useful robot data?

4. Is SOAR better at collecting useful interactions than a direct language-conditioned policy?

**Robot and task setup.** We use five WidowX 250 6-DoF robot arms in our experiments. Since we conduct long-duration autonomous data collection experiments, we installed plexiglass barriers to prevent objects from falling off the table during overnight collection. More details in Appendix C.

**Pre-training and data collection.** Before deploying SOAR, we obtain a policy that has basic manipulation skills by pre-training on BridgeData v2 [12]. We train a GCBC policy characterized by a Gaussian distribution, using a ResNet-34 as the image encoder. We use hindsight goal relabeling, and the goal images are sampled randomly from 0 to 24 future steps. Then, SOAR is deployed on 10 different scenes to autonomously collect data for $\sim$120 robot hours. Note that SOAR is able

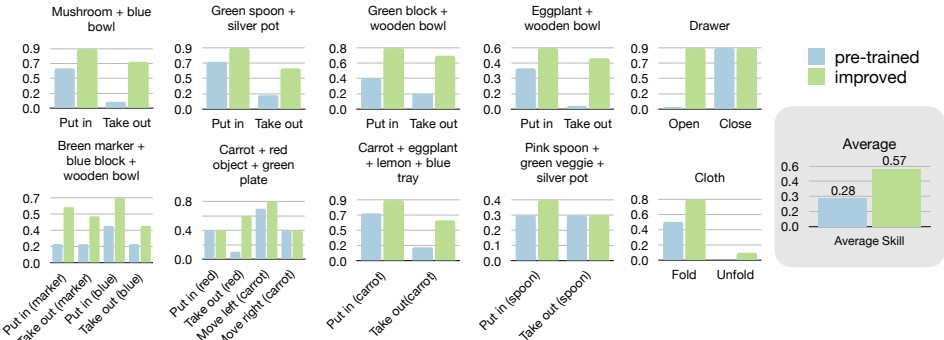

Figure 5: Autonomous improvement results in each scene: for all 10 scenes, training on scene-specific autonomous data helps to significantly improve performance over the pre-trained policy. On average, the pre-trained policy has a success rate of 28% and the improved policy has success rate 57%.

to continuously collect data without scene resets: when the policy fails to complete some VLM-proposed language task, no reset is needed to return the scene to the original condition. Instead, we use the VLM to propose a new meaningful task given the new state of the scene and objects. For the policy improvement experiments, we collected data in 10 scenes spread across 5 different table settings. All our scenes involves substantial distribution shifts from the pre-training dataset (Details in Appendix C) to test the generalization capabilities of SOAR to improve in new environments.

**Autonomous improvement with SOAR.**    After data collection, we update the policy by co-training on both the pre-training dataset (1.87M transitions) and the autonomously collected dataset (416K transitions). In Figure 5, we evaluate the decomposed language conditioned policy (GCBC + SuSIE) on the manipulation tasks from each scene, where the improved policy is co-trained on autonomous data collected only from that scene. We use the same architecture for the improvement policy as the pre-trained policy. For each scene, we evaluate $2 - 4$ skills. Across all 10 scenes, training on each per-scene autonomous dataset significantly improves the policy performance over the base pre-training dataset. On average, the success rate more than doubled the pre-training performance, jumping from 28% to 57%. Qualitatively, we observe that the improved policy is much better at manipulating objects in hard-to-grasp positions and unseen objects in the pre-training dataset.

We also trained a generalist policy using autonomous data from all 10 of the collection scenes, also with co-training on the pre-training dataset. We evaluate this generalist policy on three different scenes in Table 1. While in some scenes the generalist policy achieves the same or slightly lower success rates, on average it achieves better performance (65%) than the improved GCBC policy that is trained only on data from that scene (58%). This shows that training on more autonomous data using SOAR can bring about better improvement.

**Comparison with language-conditioned behavior cloning.**    Is decomposing language skills to language-conditioned subgoal generation and goal-conditioned control better at self-improvement than learning a direct language conditioned policy from the autonomous data? To test this, we compare against swapping the policy in SOAR with language-conditioned behavioral cloning (LCBC). We train the LCBC policy on the same autonomous data as the SOAR policy, with the language labels for LCBC coming from the VLM task proposer, and report the improvement performance on manipulation skills from six tasks across three different scenes in Table 1. We compare the two approaches trained with three different data types: (1) only pre-training data, (2) pre-training data and scene-specific autonomous data, and (3) pre-training data and all autonomous data across 10 scenes. Table 1 shows that for all six tasks redthat are unseen during pre-training, the pre-trained LCBC policy is mostly unable to achieve the task because it suffers from grounding the unseen language command to unseen objects. When trained on +scene autonomous data, LCBC policies are able to improve because the autonomous data provides such grounding. However, we find that SOAR improves much more effectively than LCBC for both autonomous dataset types. Furthermore, when trained on more autonomous data (+all), SOAR is able to improve better while LCBC performed worse. We attribute SOAR's success to using goal-conditioned policy learning for improvement: it provides a dense self-supervision and is robust to sub-optimal data. For instance, if a robot failed to "open the drawer" but the VLM success detector incorrectly marked the attempt as successful, LCBC will use this failed attempt to train the "open the drawer" policy, which ultimately degrades

| Tasks (scene #) | GCBC + SuSIE | | | LCBC | | |
|---|---|---|---|---|---|---|
| | pre-trained | +scene | +all | pre-trained | +scene | +all |
| Put green block in (#1) | 0.4 | **0.8** | 0.7 | 0.0 | 0.5 | 0.3 |
| Take green block out (#1) | 0.2 | **0.7** | 0.6 | 0.0 | 0.4 | 0.3 |
| Put carrot in (#8) | 0.4 | **0.7** | 0.6 | 0.0 | 0.4 | 0.3 |
| Take carrot out (#8) | 0.4 | 0.6 | **0.7** | 0.0 | 0.1 | 0.3 |
| Put spoon in (#9) | 0.3 | 0.4 | **0.7** | 0.0 | 0.5 | 0.5 |
| Take spoon out (#9) | 0.3 | 0.3 | **0.6** | 0.0 | 0.0 | 0.0 |
| Average | 0.33 | 0.58 | **0.65** | 0.0 | 0.32 | 0.28 |

Table 1: The decomposed language-conditioned policy in SOAR is better at autonomous improvment than LCBC. Compared to LCBC, GCBC+SuSIE is much better at utilizing autonomous data for improvement, with training on all the autonomous data showing positive transfer.

its performance. SOAR offers a more robust solution: it does not use the VLM success/language labels and improves a goal-conditioned policy through hindsight relabeling.

**Comparison with relevant methods** To evaluate the performance of SOAR, we compare it against two relevant methods: DIAL [84] and RoboFuME [19]. DIAL improves a language-conditioned policy by labeling an unlabeled human expert demonstration dataset with a VLM, though it relies on expert demonstration data to finetune the VLM rather than autonomous data. RoboFuME employs online RL to improve language skills from autonomous data but requires additional expert demonstrations to fine-tune the policy before improvement. Implementation details of both methods can be found in Appendix F. Even though these two methods both require expert demonstration data, we re-purpose them to our setting by replacing the expert data with the autonomous data SOAR collected. We test all methods on two tasks in scene 1. Our results, presented in Table 2, show that SOAR significantly outperforms both baselines. The baselines encountered challenges when learning from autonomously collected suboptimal data. We hypothesize that Robo-FuME may have struggled due to difficulties in learning a robust language-conditioned Q function on a large pre-training dataset, while DIAL's performance is constrained because it cannot effectively utilize sub-optimal autonomous data. More details and a discussion are provided in Appendix F.

| Tasks | SOAR | RoboFuME | DIAL |
|---|---|---|---|
| Put green block in (#1) | **0.8** | 0 | 0 |
| Take green block out (#1) | **0.7** | 0 | 0 |

Table 2: SOAR significantly outperforms both RoboFuME and DIAL, which heavily relies on expert demonstrations and were not able to improve with autonomous data.

**Dataset details.** Besides evaluating autonomous improvement with our system, a secondary contribution of our work is to provide a publicly available dataset of autonomously gathered robotic experience that can be used for future research on self-improvement. This dataset, SOAR-Data, consists partly of data collected during the autonomous improvement experiments with our GCBC policy on the 10 scenes in Figure 4, and also includes autonomous data collected by SOAR on various other scenes. In total, SOAR-Data has more than $30,582$ trajectories (3M transitions) collected with 53 different sets of objects across 5 different table top setups. Each trajectory in SOAR-Data comes with language annotations, 5 commanded subgoal images generated by SuSIE, and a task success label predicted by the VLM. More details can be found in Apendix E. The mixed quality nature of this dataset makes it potentially a good resource for offline reinforcement learning research.

## 5 Conclusion

In this paper, we propose SOAR, a robotic system capable of fully autonomous large scale data collection in the real world, which can use that data to improve a multitask instruction-following policy via self-supervision to 2x the pre-training performance. We find that language-conditioned skills can be effectively decomposed into a language-conditioned subgoal image generator and an image-goal conditioned policy, and such decomposition can make use of Internet-scale pre-training for semantic understanding and improve a low-level control policy with unlabeled autonomous data. We release the large autonomous dataset collected by SOAR, and in doing so demonstrate that autonomous data collection is a viable way of scaling robotic improvement, potentially even beyond the limits of human teleoperation.

**Acknowledgments**

We would like to thank Kyle Stachowicz, Aviral Kumar, Seohong Park, Kevin Black, and Mitsuhiko Nakamoto for valuable advice and discussions. This research is partly supported by NSF FRR IIS-2150826, as well as ONR N00014-20-1-2383, N00014-21-1-2838, and N00014-22-1-2773. We thank the Google TPU Research Cloud (TRC) program for their donation of TPU computing resources. Pranav is supported by the NSF Graduate Research Fellowship.

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

# A Illustrations of SuSIE-Generated Sub-goals

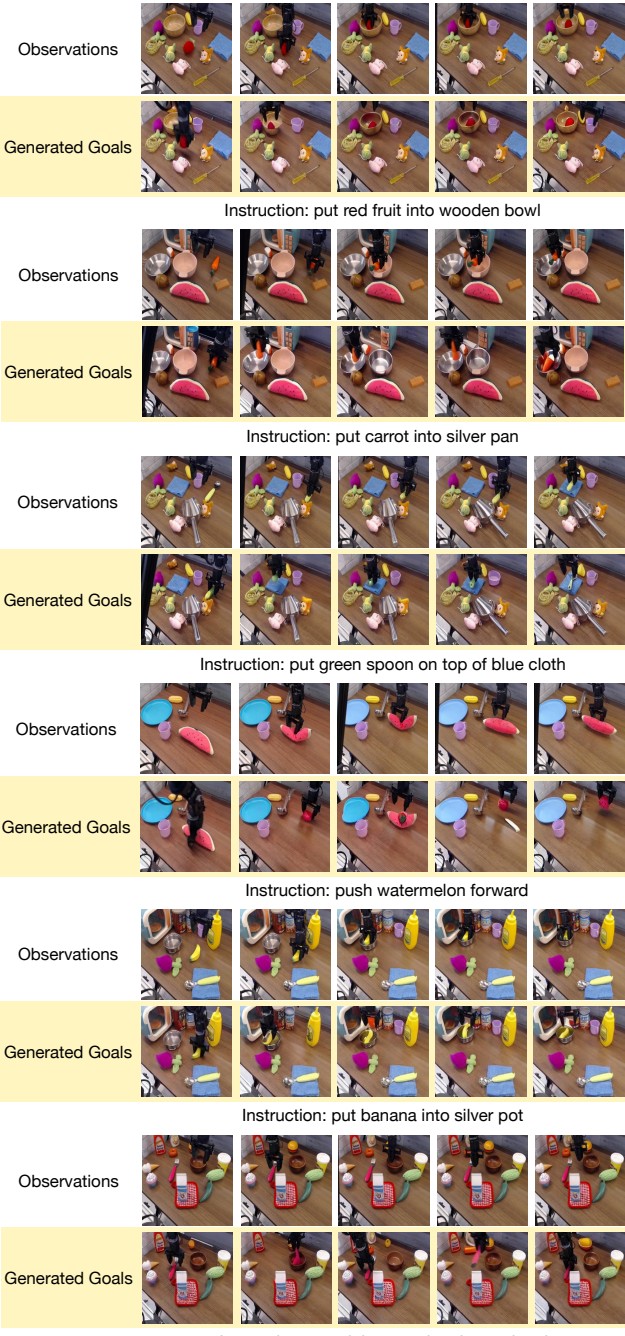

Figure 6: SuSIE generated subgoals in cluttered scenes. The observations and subgoals are collected by rolling out the SOAR policy for 100 timesteps, with a new subgoal generated every 20 timesteps (totaling 5 generated subgoals per trajectory). The subgoals are highlighted in yellow.

Here we provide some visual examples of the image sub-goals generated by the image-editing diffusion model SuSIE [23]. We show SuSIE goals in especially cluttered scenes to show its generation capabilities. All illustrations in Figure 6 is collected by rolling out the SOAR policy with the respective language instructions, and collect the five subgoals that are 20 steps apart. We find that overall the SuSIE generated goals are accurate and good for low-level control; we did observe hallucination at times, but did not find it to impact SOAR's ability to collect meaningful data.

# B    Implementation Details

## B.1    VLM Task Proposals & Success Detection

### B.1.1    Task Proposals

The task proposal problem can be defined formally as a mapping from the space of images of the robot's environment $I$ to the space of language tasks $T$. So as to account for the current capabilities of open-source VLMs and the limitation on tasks that can actually be physically completed by the robot, we strict $T$ to be a discrete set of tasks for each environment: $T = \{\tau_1, \tau_2, ..., \tau_{|T|}\}$. In our setup, task proposal amounts to determining if the language task has succeeded across $T$: we determine the success of all tasks in $T$ and use the non-successful tasks as the set of feasible tasks.

We leverage CogVLM [22], an open-source VLM, for both task proposals and success detection. Similar to many open-source VLMs, CogVLM has shown strong performance on popular image-understanding benchmarks, particularly Visual Question Answering (VQA) benchmarks [85, 86, 87, 88, 89]. We exploit the model's proficiency in VQA tasks by reframing the problem of task proposal into a VQA question. In other words, we can translate each language task into a question about the robot workspace and a boolean answer that indicates whether the task is feasible. Table 3 shows some examples of how the feasibility of a language task can be formulated as a VQA-style pair:

| Language Task $T$ | VQA Question-Answer Pair |
|---|---|
| **Original:** move the orange crayon from the blue plate to the table | **VQA-style:** Is the orange crayon currently on the blue plate? 

 **Answer that implies task is feasible:** True |
| **Original:** put the orange crayon on the cloth | **VQA-style:** Is the orange crayon on top of the cloth? 

 **Answer that implies task is feasible:** False |
| **Original:** move the red object from the blue plate to the table | **VQA-style:** Is the red object on the blue plate? 

 **Answer that implies task is feasible:** True |

Table 3: Task Proposals to VQA

This translation from language task to VQA prompt can be achieved automatically with VLM and LLMs. First, we few-shot prompt an LLM (GPT-4) to convert the language task strings from $T$ into a VQA-style question and answer pair. The specific prompts we use for the LLM is released along with our code release. Then, we feed the VQA *question* to CogVLM, alongside the initial image observation of the robot workspace before the task is attempted. Since the VLM returns a description of the workspace along with the answer, we decode the VLM reponse with an LLM (GPT-4) into a single boolean variable. This boolean varaible indicates the VLM's understanding of the VQA question in the robot workspace. Finally, task feasibility is determined by matching the boolean variable with the VQA *answer* that implies task feasibility, which is illustrated on the right column of Table 3. This process constructs the set of feasible tasks $T_{\text{feasible}} \subseteq T$, after which the upper-confidence bound ranking procedure described in Section 3 is used to select a task.

To further aid understanding, we provide a complete example of the whole process in Table 4, which shows how the task "put the green block into the brown bowl" is deemed feasible by the VLM.

| Language Task | Put the green block into the brown bowl. |
|---|---|
| Workspace initial view | |
| VQA Pair | Question:
Is the green block in the brown bowl?

Answer that implies task is feasible: False |
| VLM Output | No, the green block is placed outside the bowl, on the edge of the transparent platform. |
| LLM Decoded Output | False |
| Task feasibility | Feasible |

Table 4: Full example of how the VLM determines the feasibility of one task.

### B.1.2 Success Detection

We use the same framework as in task proposals to detect whether tasks have been successfully completed. Given a task description and the *final* view of the robot workspace when attempting the task, we can similarly translate the task into a VQA question and query the VLM for an answer. Note that often times we can use the same VQA question from task proposal for success detection, but the VQA answer that impliess success is the opposite of the VQA answer that implies a task is feasible. This is because the task is only feasible to attempt when it has not already been successfully completed.

## B.2 Goal-conditioned Policy

### B.2.1 Model Architecture

Here we outline the neural network architecture and training details of our goal-conditioned policy. The image of the current observation and desired goal observation are frame stacked along the channel dimension and fed through a ResNet-34 [90]. Instead of the usual BatchNorm [91] we utilize GroupNorm [92] in the ResNet. Following the ResNet is a 3-layer MLP which outputs the mean and standard deviation parameterizing a Gaussian action distribution. Each hidden layer has dimension 256 and uses Swish [93] activations. In practice we output just the mean (and fix the standard deviation to be state-independent).

### B.2.2 Training

To pre-train a base goal-conditioned policy on BridgeData v2 we use the Adam [94] optimizer with cosine learning rate decay from an initial $0.0003$ to $0$ over the course of $500k$ gradient steps. We also use linear learning rate warmup for 2000 gradient steps. We use a L2 weight decay of $0.001$ and for Adam use $\beta_1 = 0.9$ and $\beta_2 = 0.98$, along with clipping gradients to have maximum norm of $1.0$. Before channel-wise concatenation the current and goal images are processed via a standard series of image augmentations, including random resized cropping, brightness, contrast, saturation, and hue augmentations. During pre-training, goal images are sampled uniformly at random from $0$ to $24$ timesteps into the future, approximately matching the subgoal horizon the image subgoal generator SuSIE was trained with.

To train an improved GC-policy leveraging the autonomous data, we co-train on the autonomous data and pre-training dataset (BridgeData v2). For the improved policies trained on individual scenes, we up-sampled the autonomous data $10\times$ with respect to its proportion to the pre-training data. (e.g., When the per-scene autonomous data is the size of $3\%$ of the pre-training dataset, we use a sampling ratio of $30\%$ for the autonomous data and $70\%$ for the pre-training data. For the

generalist GC-policy trained on all the autonomous data, the dataset sampling ratio is $80\%$ for the pre-training data and $20\%$ for autonomous data. Following the approach used in BC-Zero [62], data from the autonomous data is relabeled with actions being the sum of two consecutive actions $(a'_t \leftarrow a_t + a_{t+1})$. This counteracts the tendency of the robot during autonomous data collection to take lower magnitude actions than those in the pre-training dataset, a behavior that results from the Gaussian MLP head averaging modes in the state-conditioned action distribution. Like during the pre-training stage goals are sampled uniformly at random $[0, 24]$ timesteps into the future for the pre-training data, and for the autonomous data goals are sampled $[0, 12]$ timesteps into the future. All policies are trained with a batch size of $256$.

## B.3 Language-conditioned Policy

### B.3.1 Model Architecture

The language-conditioned policy architecture is a ResNet-34 with FiLM conditioning. Language instructions are first encoded by a frozen MUSE encoder and then passed through two fully connected layers. The image observation is passed through the ResNet which is conditioned on the language embedding via FiLM layers applied at the end of every ResNet block. The MLP action head is the same for the language-conditioned policy as the goal-conditioned policy.

### B.3.2 Training

The training procedure of the language-conditioned policy is exactly the same as that for the goal-conditioned policy.

## B.4 Image Subgoal Generation

We leverage SuSIE [23] as our language-conditioned image subgoal generator. During generation we use classifier-free guidance with a weight of $2.0$ for the image and $7.5$ for the text prompt. Each autonomous trajectory consists of 5 subgoal generations with the low-level policy given 20 timesteps (identical to the horizon SuSIE was trained with) to reach the subgoal.

Generating a sequence of subgoal images one after another offers various practical advantages over creating a single final goal image. Particularly for autonomous robot deployment, a significant benefit arises when the robot's actions alter the environment in a manner unrelated to the intended goal. In such cases, iterative regeneration can seamlessly integrate these environmental changes into the generated subgoals, eliminating the necessity for the policy to reset the environment to achieve the desired goal image.

## C Robot Setups & Scene Descriptions

We use delta end-effector control with a frequency of 5 Hz. We use an RGB camera to capture the top-down third-person view of the robot workspace, and use $256 \times 256$ images as input to our control policy.

Table 5 lists the scenes, associated language tasks, and the number of successful autonomous trajectories collected for each of the 10 scenes. In each scene, we decide the objects to be placed in the scene and specify a list of meaningful language tasks, and SOAR autonomously proposes the tasks to self-practice. All scenes include **distribution shift** from the pre-training dataset. These distribution shifts include the presence of unseen objects, out-of-distribution camera viewpoints, and unseen robot environments. The inclusion of plexiglass barriers also introduces additional visual distribution shift, as no plexiglass barriers were included in the pre-training data. Specifically, two table settings are included in the pre-training dataset [12], while the other three tabletops are **unseen**. We use 19 different objects, with 14 of them **not seen** in the pre-training dataset. Such distribution shifts tests SOAR's ability to generalize and improve in new scenes.

| Scene # | Workspace image | Task Descriptions | # Autonomous Successful Trajectories |
|---|---|---|---|
| 1 |  | **1.** put the green block in the wooden bowl
**2.** remove the green block from inside the wooden bowl and put it on the table
**3.** put the red fruit in the wooden bowl
**4.** remove the red fruit from inside the wooden bowl and put it on the table | 2056 |
| 2 |  | **1.** put the purple eggplant in the brown bowl
**2.** remove the purple eggplant from inside the brown bowl and put it on the table | 282 |
| 3 |  | **1.** move the green marker to the left side
**2.** move the green marker to the right side
**3.** put the blue block in the wooden bowl
**4.** remove the blue block from inside the wooden bowl and put it on the table
**5.** put the lemon in the wooden bowl
**6.** remove the lemon from inside the wooden bowl and put it on the table | 364 |
| 4 |  | **1.** put the red object on the green plate
**2.** take the red object out of the green plate and put it on the table
**3.** put the carrot on the green plate
**4.** take the carrot out of the green plate and put it on the table | 206 |
| 5 |  | **1.** open the drawer
**2.** close the drawer | 221 |
| 6 |  | **1.** put the mushroom in the blue bowl
**2.** remove the mushroom from the blue bowl and put it on the table | 46 |
| 7 |  | **1.** put the mushroom in the metal pot
**2.** remove the mushroom from the metal pot and put it on the table
**3.** move the green spoon to the left
**4.** move the green spoon to the right | 82 |
| 8 |  | **1.** put the carrot on the blue plate
**2.** remove the carrot from the blue plate and put it on the table
**3.** put the purple eggplant on the blue plate
**4.** remove the purple eggplant from the blue plate and put it on the table
**5.** put the lemon on the blue plate
**6.** remove the lemon from the blue plate and put it on the table | 700 |
| 9 |  | **1.** put the green veggie on the blue plate
**2.** remove the green veggie from the blue plate and put it on the table
**3.** put the pink spoon on the blue plate
**4.** remove the pink spoon from the blue plate and put it on the table | 200 |
| 10 |  | **1.** fold the cloth from right to left
**2.** unfold the cloth from left to right | 523 |

Table 5: Details on 10 robot scenes

## D   Autonomous data quality.

Figure 3 shows an example of the language tasks proposed by SOAR, and the corresponding generated goal-image during an autonomous trajectory rollout. To assess the accuracy of the VLM success detector, we hand-labeled the ground truth success trajectories for a small subset (546 trajectories) of the autonomous data, and found that the VLM produces correct outcome success classifications on 78.87% of the episodes, with a precision of 0.63 and a recall of 1.0. This indicates that the VLM is very good at avoiding false negative labels, but does include a portion of false positive success labels. Despite such imperfect supervision, the SOAR policy is still able to improve dramatically. This again highlights the importance of decomposing the language-conditioned policy and using hindsight goals to learn from sub-optimal data.

Finally, we investigate whether SOAR is better at autonomous data collection compared to a language-conditioned policy. For a fair comparison, we trained a language-conditioned behavioral cloning (LCBC) policy on the same pre-training dataset, and task it to collect data autonomously, following language instructions given by the same VLM. We find that when collecting data for the mushroom + blue bowl task, which is seen in the pre-training data, LCBC achieves a 13.9% data collection success rate over the course of two hours of data collection. That is, 13.9% of the collected trajectories are successful according to the VLM. In comparison, data collection with SOAR (with a GCBC policy) achieves a 35.0% VLM success rate. However, when collecting data on an unseen table and a set of unseen objects, LCBC generally does not exhibit meaningful behavior, obtaining 0% success rate over two hours. This is consistent with prior works, which found LCBC to generalize worse to unseen objects because of grounding issues [23]. In comparison, SOAR can handle such unseen objects with 26.1% success rate, because the image-editing diffusion model is trained on Internet-scale data and can generalize to a wider category of objects. In general, LCBC methods trained only on small robotic datasets suffer from poor language grounding because their pre-training datasets do not contain very diverse objects.

## E   SOAR-Data Details

In total, SOAR-Data has $30,582$ trajectories, with $10,018$ successful trajectories and $20,564$ failure trajectories. Each trajectory is 100 steps long. SOAR-Data is collected with 53 different sets of objects across 5 different table top setups. Each trajectory in SOAR-Data comes with language annotations (from a VLM), 5 commanded subgoal images generated by SuSIE during one episode, and a task success label predicted by the VLM.

| Dataset | # Traj. | # Env. | Lang. | Failed Traj. | Public | Collection |
|---|---|---|---|---|---|---|
| RoboNet [68] | 162k | 10 | × | × | ✓ | scripted |
| MT-Opt [16] | 800k | 1 | × | ✓ | ✓ | scripted, learned |
| RGB Stacking [71] | 400k | 5 | × | ✓ | × | learned |
| BridgeData V2 [12] | 60.1k | 24 | ✓ | × | ✓ | human, scripted |
| RobotSet [95] | 98.5k | 11 | ✓ | × | ✓ | human, scripted |
| **SOAR-Data** | 30.5k | 5 | ✓ | ✓ | ✓ | 100% autonomous |

Table 6: SOAR-Data is a large and publicly available robotic manipulation dataset that is collected fully autonomously and includes both successful and failed trajectories. It has diverse scenes and all trajectories have language annotations. Uniquely among datasets containing autonomous data, the SOAR-Data setup is cheap and replicable, making it an appealing real-world benchmark for learning from sub-optimal data.

## F   Baselines Implementation and Discussion

For RoboFuME [19], the original implementation uses a small subset of Bridge Data v2 as the pre-training dataset and a four-layer CNN as the actor-critic encoder. In comparsion, SOAR uses all of Bridge Data v2 and the pre-training dataset and a ResNet-34 as the vision encoder. To make the comparison fair, we use SOAR's pre-training data and encoder architecture to implement RoboFuME. We also does not include any human demonstration data for the target task as a fair comparision to SOAR. However, as we reported in Section 4, RoboFuME was not able to successfully complete

the evaluation task after pre-training. Specifically, the learned policy was not able to pick up objects, and suffered from early grasping and imprecise gripper positioning issues. We tried tuning the conservatism parameter $\alpha$ in RoboFuME (in the underlying RL algorithm CalQL), but neither of $\alpha \in \{1, 5\}$ has non-zero performance. Furthermore, following Kumar et al. [58], we tried modifying the Q function architecture to include actions as input in every layer of the MLP, but did not observe a meaningful difference. We also applied the success reward to the last 3 steps of the trajectory, and tried terminal reward of either 10 or 0 with a step penalty of $-1$, but all variants achieved 0 success rate. We suspect this is because RoboFuME's Q function is quite brittle and does not generalize well when trained on the entirety of Bridge Data v2.

For DIAL, we finetuned a CLIP model on language annotated robot data (which we obtained from the pre-training dataset, Bridge Data V2). Via the CLIP objective we trained the model to map start and end images of trajectories to a shared representation space with the associated language instruction. This setup is identical to the setup used in DIAL. With this finetuned CLIP model, we annotate all of the autonomous data collected by SOAR on scene 1 with synthetic language annotations, following the methodology from the DIAL paper, and subsequently co-train an LCBC policy on the pre-training dataset and this synthetically annotated autonomous dataset. This LCBC policy has the same architecture as the LCBC baseline used in our experiments. Table 2 depicts the performance of DIAL compared to RoboFuME and SOAR. We found the performance of this baseline to be quite poor, which we determined was due to (1) the reliance of LCBC on high-quality language annotations, and (2) the inability of DIAL to produce the necessary high-quality annotations.

## G    Failures of Possible Modules in SOAR

First, we address how the failures of the three modules impact SOAR: (1) Occasionally we observed that the VLM task proposer failed to understand the scene and determines no tasks are viable. We resolve this by commanding a random task for the system to continue. We observe that this usually perturbs the scene and sets up the VLM for a successful retry. (2) Occasionally we observe that SuSIE failed to generate a good subgoal image due to hallucination. However, this usually only lasts for only a single subgoal image and the next subgoal, which is only 20 steps away, correctly re-directs the low-level policy. (3) Sometimes the VLM success detector incorrectly classifies the success, as described in Section 4. However, we like to note that both VLM failures do not impact SOAR's ability to self-improve. We use task proposal to guide autonomous data collection towards semantically meaningful behaviors, and success detection to bias improvement data towards semantically meaningful trajectories. However, SOAR improves with a goal-conditioned objective and does not use the VLM-generated language labels and success labels. In the case that either VLM modules fail, the collected robot trajectory is still valid and SOAR can still improve by learning to reach hindsight goals. In contrast, LCBC relies heavily on the accuracy of both the language label and success label.

Second, when the WidowX arms experience a motor failure, we immediately detect it, perform a software reboot, and restart a new trajectory in data collection. Note that over several weeks of data collection, we have found only the robot hardware failure is common. This is a limitation of the affordable robot setup. However, because of our software solution, it does not impact the data quality because all trajectories with robot failures are ignored.

## H    Limitations

Here we highlight some limitations of our work which suggest promising directions for future research. Although SOAR effectively harnesses autonomous data to improve policies significantly, further improvement could be achieved by incorporating unsuccessful autonomous trajectories as training data. Additionally, while our results showcase the capacity of the SOAR framework to robustify existing skills on unseen environments, an interesting area of future work is acquiring skills not present in the pre-training dataset through devising strategies to explore and gather data conducive to learning these new skills. Finally it would be of interest to see whether the autonomous improvement approach presented in this work could scale to dynamic tasks (e.g., throwing, pouring, wiping) or dextrous tasks (e.g., in-hand re-orientation).

