# OpenReview forum: "Autonomous Improvement of Instruction Following Skills via Foundation Models"
_robot-learning.org/CoRL/2024/Conference — CoRL 2024_

### Official Review · Reviewer_zNvs · 2024-07-13
**interesting idea with lots of questions to be addressed**

**Originality:** 3
**Technical Quality:** 2
**Clarity Of Presentation:** 3
**Potential Impact:** 3
**Recommendation:** 3
**Confidence:** 4

**Review:**

This paper presents an interesting approach to autonomous robotic learning, with some promising initial results. The use of foundation models to guide task proposal and evaluation is a clever way to leverage large-scale pre-training. The decomposition of language instructions into image subgoals also seems beneficial for learning from autonomous data.

However, there are several limitations and concerns:
1. Task Complexity: The demonstrated tasks (simple pick-and-place, drawer opening/closing) are relatively basic. It's unclear how well this approach would scale to more complex manipulation tasks or dynamic environments. The paper does not adequately address potential limitations of the image-editing diffusion model for proposing subgoals in cluttered or dynamic scenes.

2. Visual Grounding: Even state-of-the-art vision-language models like GPT-4V struggle with recognizing precise spatial relationships and object interactions. The paper doesn't sufficiently discuss how SOAR might handle these limitations for more complex instructions or scenes.

3. Task Horizon: There's limited discussion on how SOAR performs as task complexity and horizon increase. Does performance degrade for longer-horizon tasks? This is an important consideration for real-world applicability.

4. Scalability with Model Improvements: While the paper demonstrates improvement over a baseline, it's not clear how SOAR would scale as underlying models (VLMs, diffusion models) improve. Would the success detection remain reliable for more complex tasks?

5. Generalization: While the system shows some ability to handle unseen objects, more rigorous evaluation of generalization capabilities would be valuable.

**Quality Of The Limitations Section:**

1

**Questions For Rebuttal:**

1. How would SOAR handle more complex, long horizon tasks or dynamic environments?

2. What are the limitations of the image-editing diffusion model for subgoal proposal in cluttered scenes?

3. How does performance change as task horizon increases?

4. How might SOAR's performance scale as underlying foundation models improve?

=== post rebuttal
The author has clarified to some extent. I am happy to raise my score to weak accept.

**Robotics Focus:**

4

**Summary Of Paper:**

This paper presents SOAR, a system for autonomous improvement of robotic instruction-following policies without human supervision. Key components include: 1) Using a vision-language model (VLM) to propose semantically meaningful tasks, 2) Decomposing language instructions into image subgoals generated by a diffusion model and a goal-conditioned policy, 3) VLM-based success detection, and 4) Self-supervised policy improvement via goal-conditioned learning. The authors demonstrate SOAR's ability to autonomously collect data and improve performance on pick-and-place and drawer manipulation tasks across multiple scenes, including those with unseen objects.

**Summary Of Recommendation:**

The author has clarified to some extent. I am happy to raise my score to weak accept.

---

### Official Review · Reviewer_vqQn · 2024-07-18
**Leveraging Vision-Language Models for Scalable Autonomous Data Collection in Robotics**

**Originality:** 2
**Technical Quality:** 2
**Clarity Of Presentation:** 2
**Potential Impact:** 3
**Recommendation:** 2
**Confidence:** 5

**Review:**

**Strengths:**

1. Scalable Data Collection: The proposed method allows for scalable data collection, crucial for training robust robotic systems. The emphasis on autonomous and safe data collection is a significant advantage.

2. Autonomous Improvement: The system's ability to autonomously correct and improve learned behaviors without human intervention is a great contribution. This can potentially enhance the efficiency and effectiveness of robotic learning.

**Suggestions:**

- **Clarity and Typos**: Some sentences are overly long and contain typos (e.g., Lines 90-98). A thorough revision for clarity and grammatical correctness is recommended.

- **Introduction Improvement**: The introduction is vague regarding the motivation and general approach. Clarifying these aspects would enhance comprehension. Statements like "combines them in a novel way" (Line 102) are too vague. Providing specific details about the method would improve understanding.

**Overal Weaknesses:**

The paper has several notable weaknesses that need to be addressed. Firstly, there is a lack of comparison and novelty with prior works, making it difficult to evaluate the contributions relative to existing methods. The diversity and interest of the tasks proposed by the Vision-Language Model (VLM) are questionable, and there is insufficient detail on the accuracy of the generated subgoals. The VLM's limitations in reasoning about the environment further cast doubt on its effectiveness for both task generation and success detection. These items and more concerns are discussed in details in the next section.

**Quality Of The Limitations Section:**

2

**Questions For Rebuttal:**

**Weaknesses/Questions:**

1. Safety Assurance: How is safety ensured during the autonomous data collection process?

2. Diversity of Tasks: What is the diversity of tasks proposed by the Vision-Language Model (VLM)? Are the tasks sufficiently diverse and interesting?

3. Accuracy of Generated Subgoals: The authors mention training a goal-conditioned policy and using an image-editing diffusion model to generate subgoals from language tasks. How accurate are these generated subgoals?

4. VLM Limitations: The paper notes that the VLM is not sophisticated enough to fully reason about the environment and robot actions. Does this limitation restrict the diversity of tasks the robot can perform? If so, is using a VLM justified compared to hardcoding tasks?

5. Success Detection: The VLM is also used for automated success detection. Given its limitations in reasoning (as mentioned above by authors), how reliable is its success detection capability?

6. Failure Recovery: What is the failure recovery mechanism? Does the failure of a single module cause the entire trajectory to fail?

7. Environment Setup: Are the top-down RGB images in the five environments placed randomly or in specific positions relative to the robotic arm? If placement is random, how is skill transfer across environments achieved?

8. Comparison with Existing Models: How does the proposed approach compare with other VLM-based zero-shot models like Code-As-Policies and VoxPoser? Demonstrating improvements over these preexisting methods is crucial.

9. LCBC Numbers: The surprisingly low numbers of LCBC need further explanation and justification.

**Robotics Focus:**

4

**Summary Of Paper:**

The paper presents a novel approach for scalable autonomous data collection in robotics by leveraging Vision-Language Models (VLMs). The proposed pipeline uses VLMs to propose tasks, which are then executed and evaluated in combination with image generation and goal-reaching techniques, resulting in the collection of 25k trajectories across five tabletop environments. The method aims to enable safe and autonomous improvement of learned behaviors without human intervention.

**Summary Of Recommendation:**

While the proposed method of leveraging Vision-Language Models (VLMs) for scalable autonomous data collection in robotics is innovative and shows potential, there are significant weaknesses that need to be addressed. The primary concerns include the lack of comparison and novelty with prior works, questions about the diversity and interest of the tasks proposed by the VLM, and the accuracy of the generated subgoals. Additionally, the VLM's limitations in reasoning about the environment raise doubts about its effectiveness for task generation and success detection. Comparisons with existing VLM-based zero-shot models are insufficient. Addressing these concerns would significantly strengthen the paper.    **After Rebuttal**: I read the authors' rebuttal, the other reviews and the authors' responses to them. While the rebuttal and the revision has resolved some of the issues I'm still not convinced that this method is bringing improvement to the existing methods. I understand that SOAR's goal is to use a VLM to guide a pretrained policy to collect meaningful data. However, they should show that this data is actually meaningful. One of the ways of showing that this data is meaningful is to compare against the previous methods such as VoxPoser and CAP. I agree that these baselines require hand designed primitive skills but their model is also trained on 60k trajectories of Bridge Dataset, which also requires handcrafted data.

---

### Official Review · Reviewer_rBFY · 2024-07-19
**Evaluating SOAR: Advancements and Limitations in Autonomous Data Collection for Robotic Learning**

**Originality:** 3
**Technical Quality:** 2
**Clarity Of Presentation:** 3
**Potential Impact:** 3
**Recommendation:** 3
**Confidence:** 3

**Review:**

The quality of this research is commendable, presenting a well-structured approach to tackling the challenges associated with autonomous data collection and learning in robotics. The authors have conducted extensive experiments in real-world environments, effectively demonstrating the capabilities of their proposed system, SOAR. The introduction of the SOAR-Data dataset is a noteworthy contribution, offering a rich resource for future research endeavors. The paper is generally clear and well-organized, making it accessible to readers. The integration of vision-language models with autonomous data collection to enhance instruction-following skills in robots is a novel and significant advancement in the field. By enabling robots to autonomously gather and learn from data, this work addresses a critical gap where human-provided data is often expensive and limited.

**Strengths**
1. SOAR-Data: The dataset, comprising over 25,000 trajectories, serves as a valuable resource for future research.
2. Innovative Automation: The approach towards automating data collection for robotic learning tasks is a significant advancement.
3. Clarity: The writing is clear, and the explanation of the methodology is well-articulated.
4. Real-World Relevance: The experiments conducted in real-world settings enhance the practical applicability of the findings.

**Weaknesses**
1. Limited Scope of Experiments: Although the experiments are conducted in real-world settings, they primarily focus on a narrow range of skills, such as pick-and-place tasks, which may limit the generalizability of the findings.
2. Lack of Comparative Analysis: The experimental section does not include comparisons with current state-of-the-art methods that utilize vision-language models for robotic skill learning, such as "Robotic Skill Acquisition via Instruction Augmentation with Vision-Language Models" by Ted Xiao et al.

**Quality Of The Limitations Section:**

2

**Questions For Rebuttal:**

1. What are the authors' plans to broaden the range of skills and tasks that SOAR can handle?
 2. Can authors provide some quantitative comparison with existing state-of-the-art methods to show how their approach stands against these methods in terms of effectiveness and efficiency?
3. Can authors explain why the performance in Direct LCBC decrease from 32% to 28% with data from all 9 scenes but it did not in Decomposed scenario?

**Robotics Focus:**

4

**Summary Of Paper:**

SOAR is a robot learning system that autonomously improves instruction-following policies by decoupling the language-conditioned policy into an image-goal policy and a language-conditioned subgoal generator. It uses self-supervised learning, leveraging Vision-Language Models (VLMs) for task proposals and success detection.

**Summary Of Recommendation:**

The concept is intriguing, and the approach is innovative. However, the paper initially lacked comprehensive experiments and comparisons with related work. Since the authors have addressed these concerns to some extent, I recommend accepting the paper.

---

### Author Rebuttal · Authors · 2024-08-11

We thank the reviewers and the AC for their thoughtful suggestions and comments. We now provide a summary of new experiments suggested by reviewers and our response to their questions.

**New Experiments**
1. **[Reviewers rBFY, zNvs] Task Complexity:** While we acknowledge that the tasks we evaluate are not particularly dexterous or dynamic, we believe that our tasks are very much in line with other recent work [1, 2, 3, 4, 5, 6] that evaluates large-scale robotic learning. To further address this issue, we add a new experiment demonstrating SOAR’s improvement ability on a challenging deformable object manipulation task. We have updated section 4 with this new task, and include with the revised manuscript some videos depicting SOAR completing this task.
2. **[Reviewers rBFY, vqQn] Additional Baselines:** following suggestions from the reviewers, we have implemented and evaluated two new baseline approaches, DIAL [7] and RoboFuME [8], which also use vision-language models to improve policies. We find that SOAR significantly outperforms these new baselines.

In addition, we address the other concerns brought out by the reviewers, including:

3. **[Reviewers vqQn, zNvs] Subgoal Visualizations:** In Appendix A, we include visualizations of generated subgoal images by the diffusion model in various highly cluttered environments. The subgoal images remain highly coherent, handling cluttered scenes well.
4. **[Reviewers zNvs, vqQn] VLM Failures:** We address the reviewers’ concerns regarding module failures. First, we highlight that policy failure to complete a task does not require human reset. Then, we explain how possible module failures might impact SOAR, and how SOAR handles these failures.
5. **[Reviewers rBFY, vqQn] LCBC Performance:** We explain in greater detail why the LCBC baseline performs worse than SOAR.
6. **[Reviewer zNvs] Generalization Capabilities:** We clarify the degree to which our existing experiments test generalization during autonomous improvement: each of our robot setups exhibits distribution shifts in camera viewpoints, table setup, and unseen objects, providing a rigorous test for generalization.

We have included with our rebuttal response an updated manuscript, with new additions highlighted in red, and some robot videos for the new deformable manipulation task. Below we individually address the responses from each reviewer.

---

### Decision · Program_Chairs · 2024-09-04

**Decision:**

Accept

**Comment:**

Strengths:
1. SOAR leverages Vision-Language Models (VLMs) for task proposals and success detection, integrating these with an autonomous data collection framework to improve instruction-following policies.
2. The creation of the SOAR-Data dataset, with over 25,000 trajectories, is a significant contribution, providing a valuable resource for future research in robotic learning.
3. The approach towards automating data collection for robotic learning tasks addresses a critical gap in the field, reducing the dependency on human-provided data.
4. Real-world evaluations demonstrate the practical applicability of SOAR, showcasing its potential for real-world deployment.
5. The paper is generally well-written, with clear explanations of the methodology and a well-structured presentation.

Weaknesses:
1. The experiments focus primarily on pick-and-place tasks, which may limit the generalizability of the findings to more complex robotic skills.
2. The experimental section lacks comparisons with current state-of-the-art methods that utilize vision-language models for robotic skill learning, making it difficult to evaluate the novelty and effectiveness of SOAR relative to existing approaches.
3. There are concerns about the diversity and interest of the tasks proposed by the VLM, and the accuracy of the generated subgoals is not thoroughly evaluated.
4. The paper acknowledges the limitations of VLMs in reasoning about the environment and robot actions, which may affect the effectiveness of task generation and success detection.
5. There is insufficient discussion on the failure recovery mechanisms and how SOAR scales with more complex, long-horizon tasks or dynamic environments.
6. More rigorous evaluation of SOAR's ability to generalize to unseen objects and tasks is needed to fully validate its robustness and applicability.